# The Risk of Metabolic Dysfunction-Associated Steatotic Liver Disease in Moderate-to-Severe Psoriasis: A Systematic Review and Meta-Analysis

**DOI:** 10.3390/jcm14041374

**Published:** 2025-02-19

**Authors:** Suvijak Untaaveesup, Piyawat Kantagowit, Patompong Ungprasert, Nitchanan Kitlertbanchong, Tanyatorn Vajiraviroj, Tanpichcha Sutithavinkul, Gynna Techataweewan, Wongsathorn Eiumtrakul, Rinrada Threethrong, Thanaboon Chaemsupaphan, Walaiorn Pratchyapruit, Chutintorn Sriphrapradang

**Affiliations:** 1Chao Khun Paiboon Hospital, Kanchanaburi 71140, Thailand; suvijak2541@gmail.com; 2Faculty of Medicine, Chulalongkorn University, Bangkok 10330, Thailand; kantpiya@windowslive.com; 3Department of Rheumatic and Immunologic Diseases, Cleveland Clinic Foundation, Cleveland, OH 44195, USA; p.ungprasert@gmail.com; 4Department of Anatomy, Faculty of Medicine Siriraj Hospital, Mahidol University, Bangkok 10700, Thailand; nitchanan.view.89551@gmail.com (N.K.); tanpichcha.md@gmail.com (T.S.); 5Faculty of Medicine Siriraj Hospital, Mahidol University, Bangkok 10700, Thailand; nantanyatorn@gmail.com (T.V.); rinrada.thr@gmail.com (R.T.); 6College of Integrative Medicine, Dhurakij Pundit University, Bangkok 10210, Thailand; gynna.sch@gmail.com; 7Department of Medicine, Faculty of Medicine Ramathibodi Hospital, Mahidol University, Bangkok 10400, Thailand; wongsathorn.eiu@outlook.com; 8Division of Gastroenterology, Department of Medicine, Faculty of Medicine Siriraj Hospital, Mahidol University, Bangkok 10700, Thailand; ping_tnt@hotmail.com; 9Institute of Dermatology, Ministry of Public Health, Bangkok 10400, Thailand; itesatuk@gmail.com

**Keywords:** psoriasis, meta-analysis, non-alcoholic fatty liver disease, liver diseases

## Abstract

**Background/Objectives**: Psoriasis is a chronic immune-mediated skin disease associated with several metabolic comorbidities. Metabolic dysfunction-associated steatotic liver disease (MASLD) is also linked to psoriasis, but evidence regarding the severity of this association remains inconclusive. This meta-analysis aimed to investigate the relationship between MASLD and varying severities of psoriasis. **Methods**: We conducted an extensive search of four databases, MEDLINE, EMBASE, OSF, and ClinicalTrials.gov to identify relevant published articles assessing the risk of prevalent MASLD in patients with moderate-to-severe psoriasis up to April 2024. Effect estimates from each included study were combined together to calculate a pooled effect estimate for the meta-analysis using the generic inverse variance method of DerSimonian and Laird. **Results**: This meta-analysis included eight studies with a total of 109,806 participants. A 4.01-fold increased risk of prevalent MASLD was observed in patients with moderate-to-severe psoriasis compared to those without psoriasis (95% CI: 2.17, 7.77; I^2^ = 67%, *p* < 0.0001). The evidence supporting this outcome had low certainty. **Conclusions**: An incremental trend of MASLD was observed in patients with moderate-to-severe psoriasis. Routine screening for MASLD should be emphasized in this population.

## 1. Introduction

Psoriasis is a common immune-mediated skin disorder that affects both sexes and all age groups [1,2,3,4]. The increasing prevalence of psoriasis has contributed significantly to the global disease burden. For example, the incidence of psoriasis in the United States of America (USA) increased from 50.8 per 100,000 (95% CI: 41.9 to 59.6) to 88.7 per 100,000 (95% CI: 79.1 to 98.3) between 1970 and 1994 [5]. A similar trend was also observed in the Global Burden of Disease (GBD) study [6]. The prevalence of metabolic dysfunction-associated steatotic liver disease (MASLD) was incremented in moderate-to-severe psoriasis with an odds ratio (OR) of 1.45 (95% CI: 0.99–2.13) for moderate psoriasis and an OR of 1.36 (95% CI: 0.70–2.64) for severe psoriasis [7]. Moreover, studies have shown that the greater severity of psoriasis is associated with worsening quality of life, work disability, and the progression of MASLD [8,9].

The clinical manifestations of psoriasis vary depending on its type. Plaque-type psoriasis is the most common form and is characterized by symmetrically distributed, well-demarcated erythematous plaques with scales. These lesions are predominately located on the trunk, scalp, nail, and extensor areas. Other forms of psoriasis include pustular psoriasis, which presents with pustules; erythrodermic psoriasis, which is associated with widespread erythema and desquamation; and severe cases that may lead to medical emergencies [10,11]. The awareness of trigger factors, such as stress, certain medications, and skin barrier dysfunction, is crucial [11].

The concomitant diseases associated with psoriasis include cardiovascular and metabolic conditions, such as diabetes mellitus, hyperlipidemia, metabolic syndrome, and MASLD [12]. The risk of MASLD in moderate-to-severe psoriasis is primarily linked to the inflammatory process, which induces pro-inflammatory cytokines such as interleukin (IL)-6, tumor necrosis factor (TNF)-α, and leptin. These cytokines contribute to insulin resistance in psoriasis, leading to liver steatosis, fat accumulation in the liver, MASLD, and liver fibrosis [13,14]. However, insulin resistance is significantly worsened by MASLD, leading to fatty accumulation in the liver due to impaired insulin sensitivity in adipocytes [15,16]. Furthermore, cytokines from visceral fat promote inflammation and the development of MASLD. Oxidative stress resulting from fat accumulation is also proposed as a contributing mechanism [14]. MASLD can develop into liver cirrhosis and hepatocellular carcinoma, causing an impact on quality of life and poor prognosis [17].

The 2015 meta-analysis by Candia et al. primarily focused on the association between MASLD and moderate-to-severe psoriasis, compared to mild psoriasis. However, it lacked evidence on MASLD prevalence in patients without psoriasis. Furthermore, the generalizability of these findings is limited as the study included only case-control studies [18]. Similarly, Bellinato et al. conducted a meta-analysis in 2022 that assessed the risk of MASLD in chronic plaque-type moderate-to-severe psoriasis based solely on severity measured by the Psoriasis Area and Severity Index (PASI) score. As a result, concerns about generalizability arise, given the focus on only chronic plaque-type psoriasis [19]. Therefore, this systematic review and meta-analysis aims to explore the association between MASLD and psoriasis severity more comprehensively.

## 2. Materials and Methods

This systematic review and meta-analysis adhered strictly to the Preferred Reporting Items for Systematic Reviews and Meta-Analyses (PRISMA) statement, as outlined in Appendix A [20]. The protocol was registered on the International Platform of Registered Systematic Review and Meta-Analysis Protocols (INPLASY202510068) prior to execution [21].

### 2.1. Search Strategy

Eight researchers (S.U., P.K., N.K., T.V., T.S., W.E., R.T., and T.C.) independently searched four databases, MEDLINE, EMBASE, OSF, and ClinicalTrials.gov, for relevant published articles reporting on MASLD and psoriasis severity from commencement to April 2024. The unpublished articles were searched from medRxiv. We utilized keywords related to “metabolic dysfunction-associated fatty liver disease”, “non-alcoholic fatty liver disease”, and “psoriasis”, as outlined in Appendix A. References within each eligible article were meticulously reviewed to ensure the inclusion of all potential studies.

### 2.2. Inclusion and Exclusion Criteria

Eligible studies must consist of two cohorts of patients with moderate-to-severe psoriasis and individuals without psoriasis. They must compare the risk of prevalent MASLD between the two cohorts. Reported outcomes included OR, relative risk (RR), or hazard ratio (HR) with their 95% confidence interval (CI) or crude outcomes. All studies had to be published in English. To comprehensively assess the risk of MASLD, studies including patients from any and all age groups were eligible. The assessment of moderate-to-severe psoriasis severity was based on the following criteria: (1) severity assessment scores reported in the studies or guidelines, including the PASI, Dermatology Life Quality Index (DLQI), or body surface area (BSA) [22]; (2) hospitalized patients with psoriasis as the primary diagnosis; (3) diagnostic codes from any database or ICD codes [23]; (4) studies that reported moderate-to-severe severity without detailed criteria. We excluded patients undergoing systemic treatment for psoriasis, such as immunosuppressive or biologic therapies, to minimize potential confounding effects. The diagnosis of MASLD required reporting in the database [24], relevant liver function tests [25,26,27] or ultrasonography findings [9,25,26,27,28,29] with no other diagnostic methods used [30], and the exclusion of other possible causes, such as hepatotoxic drugs, Wilson disease, viral hepatitis, alcohol consumption, or liver cancer. Other metabolic comorbidities, including diabetes mellitus (DM), obesity, hypertension (HT), dyslipidemia (DLP), and metabolic syndrome, were also considered due to their role as risk factors for MASLD. Studies such as review articles, narrative reviews, systematic reviews, meta-analyses, or randomized controlled trials were excluded.

The title and abstract screening were independently evaluated based on these inclusion and exclusion criteria using Covidence, a tool for systematic review and meta-analysis screening (Covidence.org, Melbourne, VIC, Australia). Subsequently, full-text reviews were performed to identify eligible articles. Discrepancies during the screening process were resolved through discussion with other investigators (P.U., W.P., or C.S.).

### 2.3. Data Extraction

We independently extracted the following topics in the data extraction form: author, year of publication, baseline characteristics of participants including the total number of participants, mean age with standard deviation (SD), mean body mass index (BMI) with SD, number of psoriasis patients and their severity, comorbidities of participants (including DM, obesity, HT, DLP, and metabolic syndrome), criteria for severity assessment of psoriasis, and diagnostic criteria of psoriasis and MASLD. We also extracted the crude outcomes, ORs, and their 95% CIs from each included study. The other investigators (P.U., W.P., or C.S.) were consulted regarding any disparities arising during the data extraction process.

### 2.4. Quality Assessment

Two investigators (S.U. and P.K.) independently assessed the quality using the modified Newcastle–Ottawa Scale (NOS). This scale provided three domains with a total score of 10 points, encompassing selection (representativeness of the sample, sample size, non-respondents, and ascertainment of the exposure), comparability, and outcome (assessment of the outcome and statistical tests) [31]. The other investigator (P.U.) was consulted regarding any disparities arising from the quality of evidence assessment.

### 2.5. Certainty of Evidence Assessment

Two investigators (S.U. and P.K.) independently assessed the quality of evidence using the Grading of Recommendations, Assessment, Development, and Evaluation (GRADE) approach. For observational studies, the certainty of evidence was initially rated as low and could be downgraded due to the possibility of risk of bias, inconsistency, indirectness, imprecision, or publication bias [32]. The other investigator (P.U.) was consulted regarding any disparities arising from the quality of evidence assessment.

### 2.6. Statistical Analysis

We used the Review Manager 5.4 software from Cochrane to calculate pooled ORs with their 95% CIs using the generic inverse variance method of DerSimonian and Laird. [33] We computed the OR using raw data for studies without reported ORs. We used the random-effects model as the fundamental assumption of the fixed-effect model; that there is only one true effect estimate that all study should yield is almost always not true in the real world due to difference in methodology, measurement, and background population where participants were recruited/identified. Numerical results from charts were extracted using WebPlotDigitizer (automeris.io). If multiple severity assessments of psoriasis were reported in one study, we applied PASI for statistical analysis. A random-effects model was used due to the variation in baseline characteristics among participants. Heterogeneity was evaluated by Cochran’s Q test and I^2^ statistic, which were categorized as insignificant (0–25%), low (26–50%), moderate (51–75%), and high (>75%) heterogeneity [34]. We omitted the study conducted in children and adolescents for the sensitivity analysis. Subsequently, we conducted a subgroup analysis based on geographic regions (Western countries and India), PASI score for the severity assessment of moderate-to-severe psoriasis, and ultrasonography as the diagnostic criterion of MASLD.

## 3. Result

### 3.1. Search Results

Our search strategy yielded 28,558 articles (21,444 from EMBASE, 5352 from MEDLINE, 0 from OSF, 12 from ClinicalTrials.gov, and 1750 from medRxiv). After removing 4655 duplicate articles, 23,903 articles were reviewed based on their titles and abstracts. Of these, 1216 potentially relevant articles were assessed for eligibility through full text. We excluded 1208 articles due to the absence of outcomes of interest, baseline participant characteristics, types of study design, and non-English language. Hence, 8 articles were included [9,24,25,26,27,28,29,30]. Figure 1 shows the screening process and search results.

### 3.2. Baseline Participants Characteristics

Table 1 describes the baseline characteristics of the participants. A total of 109,806 participants from 8 studies were retrieved [9,24,25,26,27,28,29,30]. One study was conducted in North America [28], five studies were conducted in Asia [25,26,27,29,30], and two studies were conducted in Europe [9,24]. The method used to diagnose psoriasis varied across the studies, which included clinical examination and a diagnostic code from an administrative database (i.e., the International Classification of Diseases or ICD) [23], as delineated in Appendix A. Appendix A illustrates the severity assessment for psoriasis patients. Appendix A describes the diagnostic criteria for MASLD.

### 3.3. Quality Assessment

The quality assessment from the modified NOS in each included article generally graded in the range of 8–10 points. The median of total scores was 9 (8–10). Appendix A exhibits the quality assessment profile in each included article.

### 3.4. Certainty of Evidence Assessment

We assessed the certainty of the evidence for MASLD outcomes in moderate-to-severe psoriasis using the GRADE tool, resulting in low quality due to degradation from the risk of bias and inconsistency. The quality of evidence is displayed in Appendix A.

### 3.5. Risk of MASLD in Moderate-to-Severe Psoriasis Patients

Based on eight studies, we conducted a meta-analysis of the pooled OR to estimate the risk of MASLD in moderate-to-severe psoriasis, showing a 4.10-fold increased risk of prevalent MASLD (95% CI: 2.17, 7.77; I^2^ = 67%, *p* < 0.0001) compared with those without psoriasis [9,24,25,26,27,28,29,30]. The pooled outcome is shown in Figure 2. The funnel plot represented no publication bias, as shown in Figure 3.

### 3.6. Laboratory Parameters Associated with MASLD Outcome

Yadav et al. [27] conducted the association between inflammatory markers and psoriasis severity and found that enhancing IL-6 (*p* = 0.131 of PASI score and *p* = 0.170 for BSA) and IL-1β (*p* = 0.681 of PASI score and *p* = 0.937 for BSA) were correlated with PASI scores. Conversely, TNF-α levels were not associated with PASI scores.

### 3.7. Subgroup Analysis and Sensitivity Analysis

We performed subgroup analysis and sensitivity analyses to explore the moderate between-study heterogeneity. First, we performed a sensitivity analysis by excluding the study by Panjiyar et al. [26] from the pooled analysis, as it was the only study conducted in children and adolescents, whereas the rest were conducted in adults. Excluding this study decreased between-study heterogeneity to an almost low level (I^2^ from 67% to 52%), while the pooled result remained statistically significant (pooled OR 3.23; 95% CI 1.92–5.45) (Appendix A).

Following this sensitivity analysis, we also performed a subgroup analysis based on the geographic regions of the included studies, as it is conceivable that the risk may differ across different background populations. We calculated pooled ORs for two subgroups—studies conducted in Western countries and in India. Between-study heterogeneity decreased to a low level in the first subgroup, while the pooled result remained statistically significant (pooled OR 2.59; 95% CI 1.69–3.96; I^2^ = 29%) (Appendix A). In the second subgroup, between-study heterogeneity was also lower, and the pooled result remained statistically significant (pooled OR 4.49; 95% CI 1.37–14.68; I^2^ = 59%) (Appendix A). Utilizing the diagnostic approach of MASLD, we exclusively subgrouped using ultrasonography, the widely used noninvasive method, yielding an OR of 3.06 (95% CI 1.91–4.91; I^2^ = 30%), which diminished between-study heterogeneity while preserving statistical significance. (Appendix A) Furthermore, a subgroup analysis was adjusted according to the PASI score, the standard method for stratifying psoriasis severity in psoriasis, yielding an OR of 3.76 (95% CI 2.52–5.61; I^2^ = 36%) (Appendix A). Thus, the key sources of heterogeneity in our study were geographic regions, the diagnostic approach of MASLD, and severity assessment techniques in psoriasis.

## 4. Discussion

We provided comprehensive data on the association between MASLD and moderate-to-severe psoriasis in 8 cross-sectional studies. This meta-analysis observed a significant 4.10-fold increased risk of MASLD in patients with moderate-to-severe psoriasis.

The pathophysiology of psoriasis involves immunological systems, oxidative stress, and genetic factors [1,11,35,36]. Immunological processes play a pivotal role through both adaptive and innate immune systems. Additionally, dendritic cells and keratinocytes from the innate immune system precipitate inflammatory processes via IL-1 and TNF-α, which promote T cell differentiation [10,11]. These cytokines also induce IL-6, leading to the increased proliferation of keratinocytes in psoriasis patients [35]. Oxidative stress may stimulate keratinocyte proliferation from radical oxygen species (ROS), which also contributes to inflammation in psoriasis [36]. Moreover, the inflammatory process can lead to several metabolic comorbidities, such as insulin resistance and metabolic syndrome [37]. 

Previous meta-analyses focused the risk of MASLD in moderate-to-severe psoriasis patients only on chronic plaque psoriasis with PASI score assessment for severity, which showed a consistency with our results (adjusted HR 2.23 with 95% CI:1.73 to 2.87). This association can be explained by the accumulation of fatty acids in the liver, which is precipitated by insulin resistance, thereby inducing further insulin resistance. Furthermore, pro-inflammatory cytokines and oxidative stress contributed to insulin resistance, which facilitated the development of MASLD [19].

This meta-analysis revealed a significant association between MASLD and moderate-to-severe psoriasis, which can be explained by inflammatory and metabolic mechanisms. Inflammatory cytokines, mainly IL-6 and TNF-α, are involved in the inflammatory process of psoriasis [13]. These cytokines trigger the influx of free fatty acids into hepatocytes, leading to an overload that enhances hepatic de novo lipogenesis, converting glucose into fatty acids, while also impairing fatty acid oxidation. This results in hepatic fat accumulation [16,38]. Recent evidence has also reported significant increases in IL-6 and IL-1β across different severities of psoriasis, as measured by PASI and BSA scores [27]. Additionally, IL-23, another key cytokine in psoriasis pathogenesis, plays a role by driving an imbalance between regulatory T cells and Th17 cells, as demonstrated in in vitro study [39,40]. This imbalance promotes steatosis and steatohepatitis, further linking psoriasis severity to the development of MASLD. Furthermore, the products of oxidative stress may be linked to the severity of psoriasis and inflammatory processes [14,37]. Reactive radical oxygen species from free fatty acids contribute to insulin resistance and immune dysregulation [38]. Additionally, the increased levels of lipoprotein (a) in more severe psoriasis have been reported, which are linked to the pathophysiology of MASLD [41]. 

Our findings indicate a significantly increased risk of MASLD in patients with moderate-to-severe psoriasis. Several dermatology guidelines recommended screening of MASLD in psoriasis patients of all age groups who had metabolic comorbidities, including DM, obesity, impaired fasting glucose, and a family history of this liver disease [42,43,44]. According to gastroenterology guidelines, recommendations for the screening of MASLD in psoriasis patients are lacking [45,46]. Despite this, routine screening suggestions for patients without metabolic comorbidities or other risk factors have often been overlooked. Therefore, routine screening recommendation should be emphasized to facilitate the early management of these comorbidities. However, there is a lack of evidence regarding the cost-effectiveness of screening for these comorbidities. Therefore, further evaluation of the economic aspects should be conducted and recommended to policymakers.

The variation in baseline characteristics may be explained by geographical regions, diagnostic criteria for MASLD, and severity assessment approaches in psoriasis, as indicated by the subgroup analysis. To elucidate, Naslund Koch et al. [24] classified only moderate-to-severe cases from hospitalized patients and used ICD codes for MASLD diagnosis. Additionally, only one study, conducted by Panjiyar et al. [26], reported the risk of MASLD in children and adolescents, while the other studies focused on adults. Further research is needed to explore the applicability and ensure the robustness of our findings.

This meta-analysis demonstrated a significant increase in the risk of MASLD in patients with moderate-to-severe psoriasis. We included all types of psoriasis and severity assessment, which enhanced the generalizability of our results. Therefore, prospective cohort studies are needed to confirm our results. In addition, we strictly adhered to the PRISMA guideline to ensure transparency and reliability in our meta-analysis.

However, there are some limitations in our study. Firstly, moderate heterogeneity was observed, which comes from differences in baseline characteristics such as geographic regions, diagnostic approach of MASLD, and severity assessment techniques in psoriasis. Secondly, some details of the baseline characteristics of participants were lacking, including underlying diseases and their medical prescriptions, which may have influenced our results. Thus, our results should be interpreted with caution.

## 5. Conclusions

This meta-analysis observed a significant increase in the risk of MASLD in patients with moderate-to-severe psoriasis compared with healthy individuals without psoriasis. This finding may support routine screening recommendations of MASLD in psoriasis patients. Further prospective cohort studies are needed to explore this association.

## Figures and Tables

**Figure 1 jcm-14-01374-f001:**
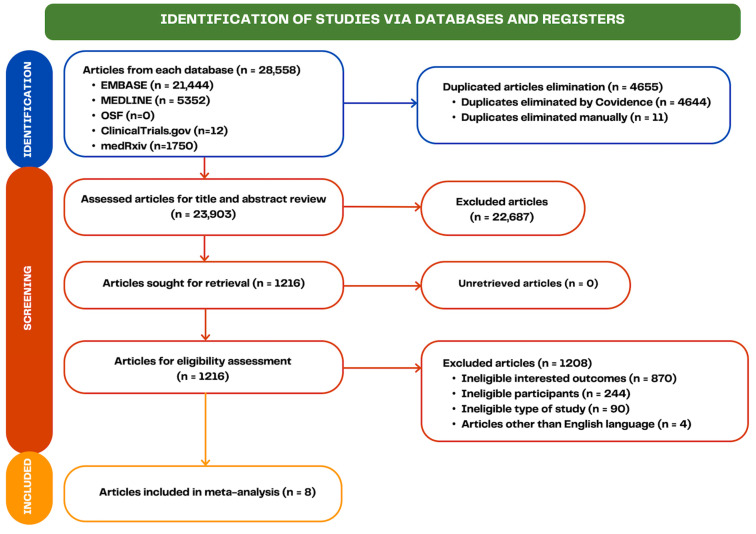
Flowchart illustrating article selection and screening procedure.

**Figure 2 jcm-14-01374-f002:**
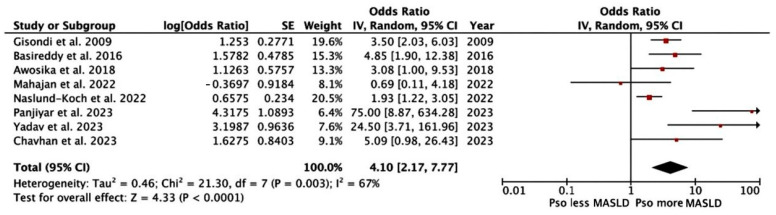
Forest plot displaying the association between MASLD and moderate-to-severe psoriasis [9,24,25,26,27,28,29,30].

**Figure 3 jcm-14-01374-f003:**
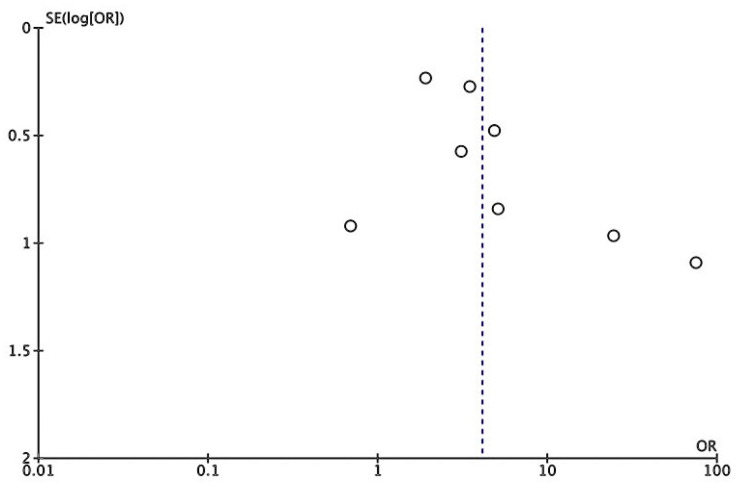
Publication bias assessing in the risk of MASLD in moderate-to-severe psoriasis.

**Table 1 jcm-14-01374-t001:** The baseline participant characteristics of the eligible articles.

References	Country	Total Participants (*n*)	Sex (Male/Female) (*n*)	Mean Age, Years (SD)	Mean BMI, kg/m^2^ (SD)	Type of Psoriasis (*n*)	Number of Psoriasis (Mild/Moderate-to-Severe) (*n*)	Number of MASLD (*n*, %)	Comorbidities (DM/HT/DLP/Obesity/MetS)	Type of Study Design	Duration of Study/Study Period
Awosika, 2018 [28]	US	152	67/85	44.37 (13.86)	27.93 (6.39)	NA	0/101	25, 16.45%	NA	CS	2009–2014
Basireddy, 2016 [30]	India	100	NA	NA	24.20 (3.67)	NA	NA/50	32, 32%	NA	CS	2012–2013
Chavhan, 2023 [25]	India	60	44/16	44.72(13.09)	NA	NA	NA/30	10, 16.67%	16 (26.67%)/21 (35%)/NA/37 (61.67%)/27 (45%)	CS	2019–2021
Gisondi, 2009 [9]	Italy	390	269/121	51.20 (10.11)	26.97 (3.95)	chronic plaque type (130)	59/71	134, 34.36%	NA/NA/NA/NA/54 (13.85%)	CS	January–June 2018
Mahajan, 2022 [29]	India	85	59/26	46.85(14.30)	NA	chronic plaque type (61)	PASI: NA/13	26, 30.59% *	NA	CS	August–December 2020
							BSA: NA/35				
Näslund-Koch, 2022 [24]	Denmark	108,835	48,936/59,899	57.69 (14.07)	25.69 (3.72)	NA	0/1277	802, 0.74%	4553 (4.18%)/48,097 (44.19%)/79,327 (72.89%)/NA/NA	CS	1997–2018
Panjiyar, 2023 [26]	India	104	64/40	11.16 (3.92)	17.55 (3.78)	chronic plaque type (36), palmoplantar psoriasis (8), sebopsoriasis (3), guttate psoriasis (2), chronic plaque type with PsA (2)	PASI: 44/8	17, 16.35%	NA/NA/16 (15.38%)/8 (7.69%)/6 (5.77%)	CS	2019–2020
							BSA: 40/12				
Yadav, 2023 [27]	India	80	52/28	38.85 (12.82)	23.67 (3.56)	chronic plaque type (50)	PASI: mild to moderate (39)/severe (11)	30, 37.5%	NA	CS	2018–2020
							BSA: mild to moderate (33)/severe (11)				

List of abbreviations: BMI, body mass index; BSA, body surface area; DLP, dyslipidemia; DM, diabetes mellitus; HT, hypertension; MASLD, metabolic dysfunction-associated steatotic liver disease; MetS, metabolic syndrome; NA, not applicable; PASI, Psoriasis Area and Severity Index; PsA, psoriatic arthritis; SD, standard deviation; US. * diagnosed by Fibroscan.

## Data Availability

The original contributions presented in the study are included in the article/Appendix A, further inquiries can be directed to the corresponding author.

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
