# Peer review of "The Risk of Metabolic Dysfunction-Associated Steatotic Liver Disease in Moderate-to-Severe Psoriasis: A Systematic Review and Meta-Analysis"

_jcm, 2025, doi:10.3390/jcm14041374_

Round 1

Reviewer 1 Report

Comments and Suggestions for Authors

I reviewed the manuscript with the title “The Risk of Metabolic Dysfunction-Associated Steatotic Liver Disease in Moderate-to-Severe 2 Psoriasis: A Systematic Review and Meta-Analysis”.

Abstract of the study is well written.

Introduction is well written.

Materials and methods: well written

Results:

Figure legends need to be more descriptive, independent to that of results section.

Discussion: well written

Conclusion: well written

Overall, well written article. Minor revisions required.

Author Response

Responses: Thank you for your suggestions. The results cannot be adequately conveyed in figure legends due to their complexity, necessitating extensive explanations.

Revision

Figure 1: Flowchart illustrating the articles selection and screening procedure

Figure 2: Forest plot displaying the association between MASLD and moderate-to-severe psoriasis

Figure 3: Publication bias assessing in the risk of MASLD in moderate-to-severe psoriasis

Reviewer 2 Report

Comments and Suggestions for Authors

Major Comments:

  1. Regarding the heterogeneity in the results section, you mention a moderate level of heterogeneity (I² = 67%). Although you used a random-effects model to address this, it is recommended to further explore the sources of heterogeneity. Could subgroup analysis or meta-regression be employed to assess how study design, geographical differences, or diagnostic criteria might influence the results? This would enhance the understanding of the sources of heterogeneity and improve the reliability of the findings.
  2. In the methods section, you mention that the diagnostic criteria for MASLD vary across different studies. It is advised to elaborate on how these differences in diagnostic criteria might impact the results in the discussion section. Would sensitivity analysis be useful to evaluate the effect of varying diagnostic criteria on the outcomes? This would strengthen the robustness of the results.
  3. The severity assessment of psoriasis varies across studies (e.g., PASI, DLQI, etc.). It is recommended to discuss how these different assessment criteria might influence the results in the discussion section. Could subgroup analysis be used to explore the impact of different severity assessment methods on the risk of MASLD?
  4. In the conclusion, you suggest enhancing MASLD screening for patients with moderate-to-severe psoriasis. It is advised to further discuss the clinical significance and feasibility of implementing this recommendation. For instance, is there enough evidence to support the widespread implementation of such screening in clinical practice? What is the cost-effectiveness of such screening? Addressing these questions in the discussion will strengthen the clinical relevance of your study.
  5. In the methods section, you mention that only MEDLINE and EMBASE databases were searched. It is recommended to consider expanding the search to include other relevant databases, such as the Cochrane Library, Web of Science, etc., to ensure that important studies have not been overlooked. Additionally, would it be possible to include grey literature or unpublished studies to minimize publication bias?
  6. A variety of search strategies can be used to retrieve unpublished studies, avoid missing a large number of negative results or small sample studies, and reduce publication bias.

  7. The future occurrence of MASLD in people with moderate to severe psoriasis and people without psoriasis can be continuously tracked, and various factors that may affect the results can be recorded, so as to clarify the causal relationship between psoriasis and MASLD, and improve the reliability of evidence.

  8. Although some possible biological mechanisms are mentioned in the article, the detailed analysis and discussion of these mechanisms are lacking, and the link between psoriasis and MASLD cannot be fully clarified.
  9. The findings show a significant increased risk of MASLD in patients with moderate to severe psoriasis, but the clinical significance and generalizability of these findings, especially applicability in different populations, have not been adequately discussed.

    Minor Comments:

    1. Line 52: ‘generic inverse variance method’ should be preceded by ‘the’.
    2. Line 53: the space in ‘Laird .’ should be removed.
    3. Line 72: the ‘a’ in ‘a worsening quality of life’ should be removed.
    4. Line 93: ‘impact’ should be preceded by ‘an’.
    5. Line 97: ‘this findings’ should be ‘thess findings’.
    6. Line 128: ‘was’ should be ‘were’.
    7. Line 128: ‘inclusion’ should be ‘inclusions’.
    8. Line 163: ‘random-effect model’ should be preceded by ‘the’.
    9. Line 164: ‘fixed-effect model’ should be preceded by ‘the’.
    10. Line 187: ‘Method’ should be ‘The method’.
    11. Line 230: 'contribute' should be ‘contributes’.
    12. Line 233: ‘consistency’ should be preceded by ‘a’.
    13. Line 256: the ‘was’ in ‘was demonstrated’ should be removed.
    14. Line 262: ‘difference’ should ‘differences’ .

Author Response

  1. Regarding the heterogeneity in the results section, you mention a moderate level of heterogeneity (I² = 67%). Although you used a random-effects model to address this, it is recommended to further explore the sources of heterogeneity. Could subgroup analysis or meta-regression be employed to assess how study design, geographical differences, or diagnostic criteria might influence the results? This would enhance the understanding of the sources of heterogeneity and improve the reliability of the findings.

Responses:

We sincerely appreciate the reviewer for raising this important concern. We have carefully considered the potential sources of heterogeneity and have conducted additional sensitivity and subgroup analyses. Our findings indicate that excluding the study conducted in the pediatric population and performing separate analyses based on geographic regions help reduce between-study heterogeneity. Furthermore, our subgroup analysis based on the diagnostic criteria of MASLD and the PASI score for psoriasis severity evaluation has further elucidated the sources of heterogeneity. Therefore, the key contributors to heterogeneity in our study include geographic regions, diagnostic approaches for MASLD, and severity assessment techniques in psoriasis.

Revision

(Materials and methods, page 8, lines 182-186)

We omitted the study conducted in children and adolescents for the sensitivity analysis. Subsequently, we conducted a subgroup analysis based on geographic regions (Western countries and India), PASI score for the severity assessment of moderate-to-severe psoriasis, and ultrasonography as the diagnostic criterion of MASLD. 

(Results, page 13, lines 244-263)

            We performed subgroup analysis and sensitivity analyses to explore the moderate between-study heterogeneity. First, we performed a sensitivity analysis by excluding the study by Panjiyar et al. from the pooled analysis, as it was the only study conducted in children and adolescents, whereas the rest were conducted in adults. Exclusion this study decreased between-study heterogeneity to an almost low level (I2 from 67% to 52%), while the pooled result remained statistically significant (pooled OR 3.23; 95% CI 1.92 – 5.45).

            Following this sensitivity analysis, we also performed a subgroup analysis based on the geographic regions of the included studies, as it is conceivable that the risk may differ across different background populations. We calculated pooled ORs for two subgroups – studies conducted in Western countries and in India. Between-study heterogeneity decreased to a low level in the first subgroup, while the pooled result remained statistically significant (pooled OR 2.59; 95% CI 1.69 – 3.96; I2=29%). In the second subgroup, between-study heterogeneity was also lower, and the pooled result remained statistically significant (pooled OR 4.49; 95% CI 1.37 – 14.68; I2=59%). Utilizing the diagnostic approach of MASLD, we exclusively subgrouped using ultrasonography, the widely used noninvasive method, yielding an OR of 3.06 (95% CI 1.91-4.91; I2 = 30%), which diminished between-study heterogeneity while preserved statistical significance. Furthermore, a subgroup analysis adjusted according to the PASI score, the standard method for stratifying psoriasis severity, yielding an OR of 3.76 (95% CI 2.52-5.61; I2 = 36%). Thus, the key sources of heterogeneity in our study were geographic regions, the diagnostic approach of MASLD, and severity assessment techniques in psoriasis.

  1. In the methods section, you mention that the diagnostic criteria for MASLD vary across different studies. It is advised to elaborate on how these differences in diagnostic criteria might impact the results in the discussion section. Would sensitivity analysis be useful to evaluate the effect of varying diagnostic criteria on the outcomes? This would strengthen the robustness of the results.

Responses:

We conducted a subgroup analysis based on the diagnostic criteria of MASLD, which revealed the cause of heterogeneity. Furthermore, we also discussed the subgroup analysis diagnostic criteria of MASLD, as per your suggestions.

Revision

(Results, page 13, lines 257-259)

Utilizing the diagnostic approach of MASLD, we exclusively subgrouped using ultrasonography, the widely used noninvasive method, yielding an OR of 3.06 (95% CI 1.91-4.91; I2 = 30%), which diminished between-study heterogeneity while preserved statistical significance.

 (Discussion, page 15, lines 307-313)

The variation in baseline characteristics may be explained by geographical regions, diagnostic criteria for MASLD, and severity assessment approaches in psoriasis, as indicated by the subgroup analysis. To elucidate, Naslund Koch et al. classified only moderate-to-severe cases from hospitalized patients and used ICD codes for MASLD diagnosis. Additionally, only one study, which was conducted by Panjiyar et al., reported the risk of MASLD in psoriasis from children and adolescents, while other studies reported it in adolescents. Further research is needed to explore the applicability and ensure the robustness of our findings.

  1. The severity assessment of psoriasis varies across studies (e.g., PASI, DLQI, etc.). It is recommended to discuss how these different assessment criteria might influence the results in the discussion section. Could subgroup analysis be used to explore the impact of different severity assessment methods on the risk of MASLD?

Responses: We conducted a subgroup analysis based on the PASI score for psoriasis severity evaluation, which revealed a source of heterogeneity. Furthermore, we also discussed the subgroup analysis of psoriasis severity assessment, as per your suggestions.

Revision (Results, page 13, lines 260-262)

Furthermore, a subgroup analysis adjusted according to the PASI score, the standard method for stratifying psoriasis severity, yielding an OR of 3.76 (95% CI 2.52-5.61; I2 = 36%).

  1. In the conclusion, you suggest enhancing MASLD screening for patients with moderate-to-severe psoriasis. It is advised to further discuss the clinical significance and feasibility of implementing this recommendation. For instance, is there enough evidence to support the widespread implementation of such screening in clinical practice? What is the cost-effectiveness of such screening? Addressing these questions in the discussion will strengthen the clinical relevance of your study.

Responses: We added a suggestion for further research in discussion section, as there no supporting cost-effectiveness studies.

Revision (Discussion, page 15, lines 304-306)

 However, there is a lack of evidence regarding the cost-effectiveness of screening for these comorbidities. Therefore, further evaluation of the economic aspects should be conducted and recommended to policymakers.

  1. In the methods section, you mention that only MEDLINE and EMBASE databases were searched. It is recommended to consider expanding the search to include other relevant databases, such as the Cochrane Library, Web of Science, etc., to ensure that important studies have not been overlooked. Additionally, would it be possible to include grey literature or unpublished studies to minimize publication bias? A variety of search strategies can be used to retrieve unpublished studies, avoid missing a large number of negative results or small sample studies, and reduce publication bias.

Responses: Thank you for your valuable suggestion. In response, we conducted searches and screenings in additional databases, including OSF and ClinicalTrials.gov. Furthermore, unpublished articles were reviewed from medRxiv. However, no additional included studies were identified through this expanded search and screening process. We have updated the results, the PRISMA flow of the selection and screening process, and Table S2 to reflect the search strategy.

Revision (Results, page 9, lines 191-193)

 Our search strategy yielded 28,558 articles (21,444 from EMBASE, 5,352 from MEDLINE, 0 from OSF, 12 from ClinicalTrials.gov, and 1,750 from medRxiv). After removing 4,655 duplicate articles, 23,903 articles were reviewed based on their titles and abstracts.

  1. The future occurrence of MASLD in people with moderate to severe psoriasis and people without psoriasis can be continuously tracked, and various factors that may affect the results can be recorded, so as to clarify the causal relationship between psoriasis and MASLD, and improve the reliability of evidence. Although some possible biological mechanisms are mentioned in the article, the detailed analysis and discussion of these mechanisms are lacking, and the link between psoriasis and MASLD cannot be fully clarified.

Responses: The biological mechanisms and further explanation of possible mechanisms were also more described in the discussion part. 

Revision (Discussion, pages 14-15, lines 282-296)

This meta-analysis revealed a significant association between MASLD and moderate-to-severe psoriasis, which can be explained by inflammatory and metabolic mechanisms. Inflammatory cytokines, mainly IL-6 and TNF- α, are involved in inflammatory process of psoriasis. These cytokines trigger the influx of free fatty acids into hepatocytes, leading to an overload that enhances hepatic de novo lipogenesis, converting glucose into fatty acids, while also impairing fatty acid oxidation. This results in hepatic fat accumulation. Recent evidence has also reported significant increases in IL-6 and IL-1β across different severities of psoriasis, as measured by PASI and BSA scores. Additionally, IL-23, another key cytokine in psoriasis pathogenesis, plays a role by driving an imbalance between regulatory T cells and Th17 cells, as demonstrated in in vitro study. This imbalance promotes steatosis and steatohepatitis, further linking psoriasis severity to the development of MASLD. Furthermore, products of oxidative stress may be linked to the severity of psoriasis and inflammatory processes. Reactive oxygen species from free fatty acids contribute to insulin resistance and immune dysregulation. Additionally, increased levels of lipoprotein (a) in more severe psoriasis have been reported, which are linked to the pathophysiology of MASLD.

  1. The findings show a significant increased risk of MASLD in patients with moderate to severe psoriasis, but the clinical significance and generalizability of these findings, especially applicability in different populations, have not been adequately discussed.

Responses: We suggested further research to explore the applicability of these findings in different populations.

 Revision (Discussion, page 15, lines 307-313)

The variation in baseline characteristics may be explained by geographical regions, diagnostic criteria for MASLD, and severity assessment approaches in psoriasis, as indicated by the subgroup analysis. To elucidate, Naslund Koch et al. classified only moderate-to-severe cases from hospitalized patients and used ICD codes for MASLD diagnosis. Additionally, only one study, conducted by Panjiyar et al., reported the risk of MASLD in children and adolescents, while the other studies focused on adults. Further research is needed to explore the applicability and ensure the robustness of our findings.

Minor Comments:

Line 52: ‘generic inverse variance method’ should be preceded by ‘the’.

Line 53: the space in ‘Laird .’ should be removed.

Line 72: the ‘a’ in ‘a worsening quality of life’ should be removed.

Line 93: ‘impact’ should be preceded by ‘an’.

Line 97: ‘this findings’ should be ‘thess findings’.

Line 128: ‘was’ should be ‘were’.

Line 128: ‘inclusion’ should be ‘inclusions’.

Line 163: ‘random-effect model’ should be preceded by ‘the’.

Line 164: ‘fixed-effect model’ should be preceded by ‘the’.

Line 187: ‘Method’ should be ‘The method’.

Line 230: 'contribute' should be ‘contributes’.

Line 233: ‘consistency’ should be preceded by ‘a’.

Line 256: the ‘was’ in ‘was demonstrated’ should be removed.

Line 262: ‘difference’ should ‘differences’ .

Responses:  We have revised the wordings as per your suggestions. However, line 128 (before revision) referred to the inclusion and exclusion criteria; therefore, we did not modify it to “inclusions.”.

Reviewer 3 Report

Comments and Suggestions for Authors

Dear Authors, the topic you describe is very interesting, very often in skin pathologies the systemic pathologies associated with them are forgotten. The structure of the manuscript is well organized, the data are well presented in the synoptic tables, the language is fluid and clear, the results are supported by encouraging data. A few questions: in the inclusion criteria, what age group was considered? What medications were the patients taking? Did the patients have comorbidities? So, to make an immediate diagnosis of MASLD, what investigations should be requested immediately?

Author Response

Responses: We described the inclusion and exclusion criteria in the Materials and Methods section, as per your suggestions. Additionally, the lack of baseline participant information, including underlying diseases and medical prescriptions from some included studies, was discussed in the limitations part.

Revision

(Materials and methods, page 6, lines 122-124)

To comprehensively assess the risk of MASLD, studies including patients from all age groups were eligible.

(Materials and methods, pages 6-7, lines 128-137)

 We excluded patients undergoing systemic treatment for psoriasis, such as immunosuppressive or biologic therapies, to minimize potential confounding effects. The diagnosis of MASLD required reporting in the database, relevant liver function tests, or ultrasonography findings, with no other diagnostic methods used, and the exclusion of other possible causes, such as hepatotoxic drugs, Wilson disease, viral hepatitis, alcohol consumption, or liver cancer. Other metabolic comorbidities, including diabetes mellitus (DM), obesity, hypertension (HT), dyslipidemia (DLP), and metabolic syndrome, were also considered due to their role as risk factors for MASLD.

(Discussion, page 16, lines 321-323)

Secondly, some details of the baseline characteristics of participants was lacking, including underlying diseases and their medical prescriptions, which may have influenced our results.

Round 2

Reviewer 2 Report

Comments and Suggestions for Authors

No more comments for the authors.